# Phase-dependent amplification of working memory content and performance

Sanne ten Oever 1,2,3✉, Peter De Weerd1,4,5 & Alexander T. Sack 1,4,6

Successful working memory performance has been related to oscillatory mechanisms operating in low-frequency ranges. Yet, their mechanistic interaction with the distributed neural activity patterns representing the content of the memorized information remains unclear. Here, we record EEG during a working memory retention interval, while a task-irrelevant, high-intensity visual impulse stimulus is presented to boost the read-out of distributed neural activity related to the content held in working memory. Decoding of this activity with a linear classifier reveals significant modulations of classification accuracy by oscillatory phase in the theta/alpha ranges at the moment of impulse presentation. Additionally, behavioral accuracy is highest at the phases showing maximized decoding accuracy. At those phases, behavioral accuracy is higher in trials with the impulse compared to no-impulse trials. This constitutes the first evidence in humans that working memory information is maximized within limited phase ranges, and that phase-selective, sensory impulse stimulation can improve working memory.

[1] Department of Cognitive Neuroscience, Faculty of Psychology and Neuroscience, Maastricht University, P.O. Box 616, 6200 MD Maastricht, The Netherlands. [2] Max Planck Institute for Psycholinguistics, P.O. Box 310, 6500 AH Nijmegen, The Netherlands. [3] Donders Centre for Cognitive Neuroimaging, P.O. Box 9010, 6500 GL Nijmegen, The Netherlands. [4] Maastricht Brain Imaging Center, 6229 EV Maastricht, The Netherlands. [5] Maastricht Centre for Systems Biology (MaCSBio), Faculty of Science and Engineering, Maastricht University, P.O. Box 616, 6200 MD Maastricht, The Netherlands. [6] Department of Psychiatry and Neuropsychology, School for Mental Health and Neuroscience (MHeNs), Brain and Nerve Centre, Maastricht University Medical Centre+ (MUMC+), P.O. Box 616, 6200 MD Maastricht, The Netherlands. ✉email: sanne.tenoever@mpi.nl

Successful working memory performance requires the maintenance of information during a retention period. Activity in the alpha (8–12 Hz) and theta (4–8 Hz) band can be observed during retention in hippocampal[1,2], sensory[3,4], and frontal regions[5,6]. Moreover, this activity has been related to successful memory performance[1,7]. However, although measuring the activity of ongoing oscillations is informative, it does not provide strong evidence that the measured activity is functionally related to the memory content being maintained. For example, the oscillatory activity could represent a cognitive control mechanism[8], or a general vehicle for neural representations[9], but not the direct representation of the specific memory content. To unravel the mechanistic role of oscillatory processing in working memory, it is paramount to isolate processes related to the neural representation of memory information itself.

Information can be represented in the distributed activity of a neural ensemble[10,11]. The strength of information representation in turn might be linked to the phase of theta and alpha oscillations. Specifically, it has been suggested that information is most strongly represented at restricted phases for which excitability levels are high[12,13]. This is corroborated by the clustering of spiking and gamma activity at restricted phase ranges in many animal recording studies[14–18] and high gamma power in human studies[19,20]. With respect to working memory, distributed activity may shows increased information content within restricted phase ranges of alpha and theta, thereby increasing the strength of the representation at specific subranges of oscillatory alpha and theta phases[9,21]. To the best of our knowledge, phase-clustered distributed activity has never been related to oscillatory fluctuations in human working memory research (see ref. [18] for animal work on this topic).

Recently, various human EEG studies have shown that after the presentation of a high-intensity impulse stimulus (either high-contrast visual stimulation or a TMS pulse) during a working memory retention interval, the memorized information can be decoded from EEG[22–24]. The success in decoding indicates the impulse stimulus' capacity to induce an amplification of the read-out of the neural memory trace. While these results illustrate improved read-out of the neuronal memory representations, no behavioral improvements related to this increased read-out were found. We expected that if the read-out of neuronal representations of memorized information is enhanced, it should be coupled with behavioral performance increases in the working memory task. Given the proposed theoretical framework suggesting that the phase of low-frequency oscillations represents working memory content[9,25], we additionally expected that the strength of distributed neural working memory representation would show oscillatory modulations, paired with corresponding oscillatory variations in behavioral measures of working memory performance.

To test these hypotheses, we analyzed EEG from 19 participants who judged a test grating against a memorized sample grating presented earlier in time (both presented for 200 ms). Here, we consider the trials in which the sample and test items were separated by an interval of 2.6 s during which the orientation of the grating had to be maintained in working memory. Midway through the retention interval, a bullseye high-contrast visual stimulus (impulse stimulation) was presented for 200 ms. We find that strength of the neural representation held in working memory as well as the accuracy of working memory performance depends on the phase of ongoing alpha and theta oscillations at which the impulse stimulation is applied. Previous human memory research has focused on univariate phase-dependent changes, demonstrating oscillatory modulations in neuronal activity strength while items are held in working memory[13,19], a result that does not allow for an inference

regarding modulations of neuronal content. Instead, the current study focuses on multivariate phase-dependent modulations of the (distributed) neural representation, allowing for inferences regarding the memory content and associated behavioral working memory performance.

## Results

**EEG decoding reveals phase-modulated information.** To test phase-dependencies of the strength of neural working memory content representation, we first operationalized the strength of information content as the multivariate classification success in decoding the specific item held in working memory. The multivariate classifier was trained on the data collected during sample item presentation to achieve maximally accurate discrimination of the sample's orientation as belonging to one of four equal 90° bins (using LDA; see Supplementary Fig. 1 and Methods). The training was performed in a time-resolved manner, using 12 ms bins centered at time points ranging between 0 and 250 ms from sample onset. Then, we tested whether the trained classifier could identify the orientation of the working memory item in 12 ms data bins collected 0–250 ms following impulse stimulus onset. In each 12 ms time bin following impulse stimulus onset, classifier performance was tested against the time-matched time bin following onset of the memory stimulus (see Supplementary Fig. 2a for the weights). The time-resolved analysis of classifier performance was necessary as we did not know in advance at which delay the effect of the impulse stimulus would affect the content information present in the EEG signal (which for example depends on information processing delays in the brain).

In addition, to relate the strength of working memory content to the phase of low-frequency oscillations, we tested the effect of the alpha and theta phase at impulse stimulus onset on time-resolved classification success of the working memory item. To test phase-dependent modulation of classifier performance in the different time bins, we first estimated phase at impulse onset (for frequencies ranging from 4 to 15 Hz) by extracting the phase from the FFT of data in a three-cycle time window preceding impulse onset (Methods). We only included data preceding the onset of the impulse stimulus to avoid effects of smearing the post-stimulus window into the phase estimation (see refs. [26,27]). Note that alpha and theta phase just prior to impulse stimulus onset can be expected to be random across trials, thus permitting tests of phase-dependent classification accuracy. Secondly, we then quantified the relationship between accuracy of classifier performance in a given time bin and phase at impulse stimulus onset. The accuracy-phase relationship was expressed by a vector calculated from the phases of the correctly classified trials. This vector has a vector angle (VA) and a vector length (VL). The VA represents the circular mean of the phase distribution. The VL is related to the width of the phase distribution and expresses how systematically a correct classification is related the VA, with values ranging from zero (no relation) to one (perfect relation; for details see Methods).

For frequencies 4–15 Hz and for each time bin, the significance of VL was evaluated using cluster statistics[28]. To this end, the observed VL was compared to the median of a baseline VL distribution generated by 1000 permutations with randomized accuracy labels, calculated per individual (Methods). Note that this analysis allows for multiple comparison correction, but not to make direct conclusions on the onset of the time and frequency point within the investigated data window[28,29]. Significant phase-dependent modulation of the decoding accuracy was present at two clusters overlapping in frequency content: (1) between 4 and 10 Hz, 0.19 and 0.22 s after impulse onset and (2) between 6.7 and 14.5 Hz, 0.13 and 0.14 s after impulse onset (see Fig. 1a; cluster

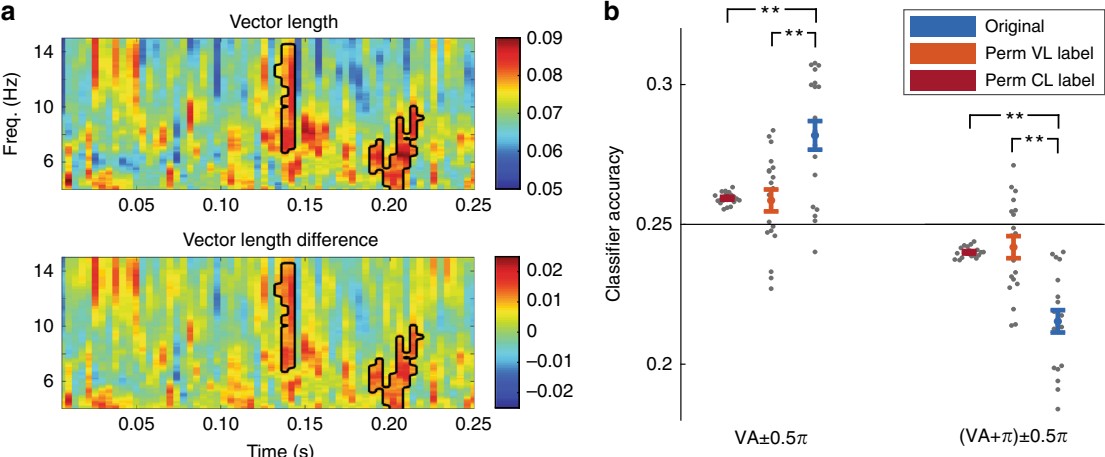

**Fig. 1 Phase-dependent decoding of working memory content by a classifier trained on the sample item. a** Vector length (VL) results shown in time frequency representations. Time zero represents impulse onset (impulse duration was 200 ms). Top figure indicates the absolute VL. The bottom figure represents the difference between the VL and the random VL based on permutations. The black contour indicates significant clusters at an alpha of 0.05. **b** Results of the generalization for the two phase bins (at the vector angle (VA) and VA + π for the frequency and time point of the maximum VL-value). Note that this relates to a post hoc analysis of the main effect of Fig. 1a. Error bars represent the standard error of the mean ($n = 19$). The original data is compared against permuted correct/incorrect label at the stage of calculating the VL (perm VL label) or permutation orientation label at the stage of calculating the classification accuracies (perm CL label). Two-tailed paired $t$-test, **$p < 0.01$.

statistics: 2.49; $p = 0.016$ and 2.03; $p = 0.03$ for cluster one and two respectively). To test for the reliability of these clusters, we correlated the VL calculated with the odd and even trials for each data point over participants. Within the cluster an average correlation of 0.168 was found, while outside of the cluster this correlation was 0.004. This indicates that theta/alpha phases at impulse onset influenced classification performance of the memory item. This is an important result, as it supports the idea that memory read-out amplification and consequent classifier performance is phase-dependent.

The analysis of VL is blind to the values of VA in individual participants. Therefore, we next explored whether the phase with best memory amplification was consistent over participants. While invasive recording data suggest a systematic relationship between phase and the activity level across individuals (see e.g.[16]), human electrophysiology has shown mixed results regarding the phase consistency across participants for pre-stimulus effects. For example, Mathewson et al.[30] have found a systematic relationship, whereas others found variations among participants[31,32]. To test across-participant consistency of phases with maximized decoding in our analysis, we extracted for per participant the VA at the maximum VL-value of the two significant clusters. For these phases, the vector showed different phase angles (VAs) across participants, with a distribution that was not significantly different from uniform (Supplementary Fig. 3; Cluster 1: Rayleigh test $Z = 0.945$, $p = 0.394$; Cluster 2: Rayleigh test $Z = 0.574$, $p = 0.566$). This indicates that although phase modulated the decoder performance for each participant, the phase of maximal decoding performance was not consistent across participants. This may be due to the fact that in EEG signal, the phase of oscillatory signals measured at the scalp is determined by many factors, for example volume conduction and the superposition of electric potentials, that could reduce this phase consistency over participants. Alternatively, the exact phase at which the most working memory information (in contrast to the most activation) is present is not identical over participants.

So far, we have shown that success of the linear classifier in extracting working memory content (as indexed by VL) was modulated by phase, but that the specific phase (VA) at which classification was optimal differed across participants. This raised

the question whether the classifier performance around the individual VA was higher than expected by chance classification performance. To quantify and test the expected gain in classifying accuracy, we extracted the mean classifier accuracy around the VA (from $-0.5\pi$ to $0.5\pi$ around the VA) and around the VA + π (from $-0.5\pi$ to $0.5\pi$ around the VA). Note that this relates to a post hoc analysis of the main effect. We maximized the bins around VA and VA + π to optimize statistical power, but in the supplementary materials we also split the bins in smaller parts (Supplementary Fig. 4a). We tested the observed classifier accuracy to the median of the chance distributions of accuracies obtained after 1000 permutations of correct/incorrect classification labels at the stage of calculating the VL. Finally, the classifier accuracy was also tested against null distributions achieved by permuting the original labels (orientation labels) going into the decoding. Initially, we entered the resulting values in a ANOVA with factors phase bin (VA and VA + π), cluster (cluster 1 and cluster 2), and permutation type (original data, permutation at VL stage, permutation at decoding stage). However, no main difference was found between the two clusters shown in Fig. 1a ($F(1,18) = 2.048$, $p = 0.17$; Supplementary Fig. 4b). Therefore, we selected the maximum VL from the pooled data over clusters for this and the following analyses. Also, no significant VL was found for higher frequency ranges or using all channels or only frontal channels (Supplementary Fig. 4c–f).

We found an accuracy of 28.2% and 21.5% at the VA and VA + π bin respectively (blue bars in Fig. 1b + Supplementary Fig. 2B for the full time course). Significant increases in the accuracy for the observed VA bin compared to the correct/incorrect label permutations were also found ($t(18) = 10.07$, $p < 0.001$; $t(18) = -9.94$, $p < 0.001$). Finally, also the classifier accuracy was significantly different from permutations based on the original orientation labels for the decoding ($t(18) = 4.35$, $p < 0.001$; $t(18) = -6.12$, $p < 0.001$). These analyses confirm that theta/alpha phase modulated the performance of the memory item decoding and that more information, i.e., a stronger neural representation, about the memorized stimulus orientation was present in a broad range of phases centered on the individual VA. Importantly, the average event-related potentials associated with the VA bin and the opposite bin did not significantly differ from each other for both training and

testing trials (Supplementary Fig. 5; cluster with lowest p-value Memory item: clusterstatistics $= -1.82$, $p = 0.112$ at 0.128–0.140 s; cluster with the lowest p-value Impulse item: clusterstatistics $= -0.81$, $p = 0.311$ at 0.136–0.140 s). This excludes that classification performance at the VA is solely due to stronger event-related responses or better signal-to-noise ratios at the VA[33]. Instead, the decoding finding can only be explained if the distributed response profile evoked by the impulse more closely matches the response profile of the memorized item when the impulse is presented within a specific restricted phase range.

**Impulse presentation at specific phases improves performance.** The previous analysis investigated whether phase influences the content representation of memorized items. Next, we investigated whether and how phase modulates behavioral working memory performance. We selected trials with orientation differences between the sample and test item corresponding to performance levels below 80%. This selection of more difficult trials was necessary because more easy trials (>80% correct performance) in which performance would already be near ceiling preclude the detection of phase-dependent performance benefits. Again, a VL analysis was performed. This time, we extracted the phase at the impulse stimulus onset for each behaviorally correct trial, weighting the trials by the orientation difference between the test and sample item (normalized weights using the mat2gray function in matlab on 34 [maximum orientation difference] minus the orientation difference; see Methods). As in Fig. 1, the present analysis was centered at the oscillation frequency showing the highest individual VL decoding modulation. Oscillatory phase at impulse onset indeed modulated the accuracies (Fig. 2a; $t(18) = 2.64$, $p = 0.008$; See Supplementary Fig. 6 for the two clusters separately; no main difference between the clusters $t(18) = 0.65$, $p = 0.524$).

The previous analysis showed that the phase at which an impulse stimulus is presented influences subsequent behavioral working memory performance. To expand on this finding, we next investigated the memory performance around the VA $(+/-\pi)$ and the VA $+ \pi$ $(+/-\pi)$, as well as comparing these performances to trials where the impulse stimulus was absent. Therefore, we extracted the weighted accuracy (weighted by difficulty, see Methods) for two phase bins and the respective permutations, similar to the decoding analysis. Behavioral accuracies at the phase bins centered on participants' VA were

significantly higher than expected from permutation-based distributions (VA bin: $t(18) = 4.09$, $p < 0.001$; VA $+ \pi$ bin: $t(18) = -4.09$, $p < 0.001$).

If neuronal working memory representations were activated by presenting an impulse stimulus at specific phase ranges, memory performance during these trials should be better compared to trials of the same length without an impulse stimulus. To test this hypothesis, we conducted a one-way ANOVA including the factor trial type with the levels VA, VA $+ \pi$ and noImpulse. This analysis showed a main effect of trial type (Fig. 2b; $F(2,36) = 17.43$, $p < 0.001$). Follow-up test showed that the accuracy at the VA trials was higher than the no impulse trials ($t(18) = 4.21$, $p < 0.001$), but the accuracy in the VA $+ \pi$ did not differ from the no impulse trials ($t(18) = -1.12$, $p = 0.276$). To ensure that this effect was not due to extracting the phases at the mean phase of the accurate trials, we did the alignment for the noImpulse trials, using random phase angles. Still, after controlling for this alignment, the VA was significantly different from the no impulse trials ($t(18) = 2.34$, $p = 0.031$).

Finally, to obtain further support for the phase-dependency of behavioral classification performance, we computed for each participant the difference between the VA from classifier data and the VA from behavioral data. This analysis revealed that the VA differences were uniformly distributed on zero as shown by V-statistics (Fig. 2c[34]; $Z(19) = 12.10$, $p < 0.001$) or by permutations of the absolute phase difference (Fig. 2d; $p = 0.017$). This demonstrates that the phases at which the behavioral accuracies and classifier accuracies were highest converged to the same phase.

The results presented here are to our knowledge the first demonstration in humans that neural memory content representation fluctuates in a phase-dependent manner at theta/alpha frequencies, and that the read-out of the neural representation as well as the corresponding working memory performance can be enhanced by a phase-specific sensory impulse stimulus. Moreover, the phase-dependent enhancements of working memory information content as quantified here by decoder accuracy were mirrored by corresponding enhancements of human working memory performance. These findings show for the first time the interaction between oscillation phase and information content during working memory as postulated in some theoretical models (e.g.,[9,35]). Remarkably, this interaction was also present in publicly available EEG data from Wolff et al.[23], who in their study focused exclusively on the successful decoding of working memory content

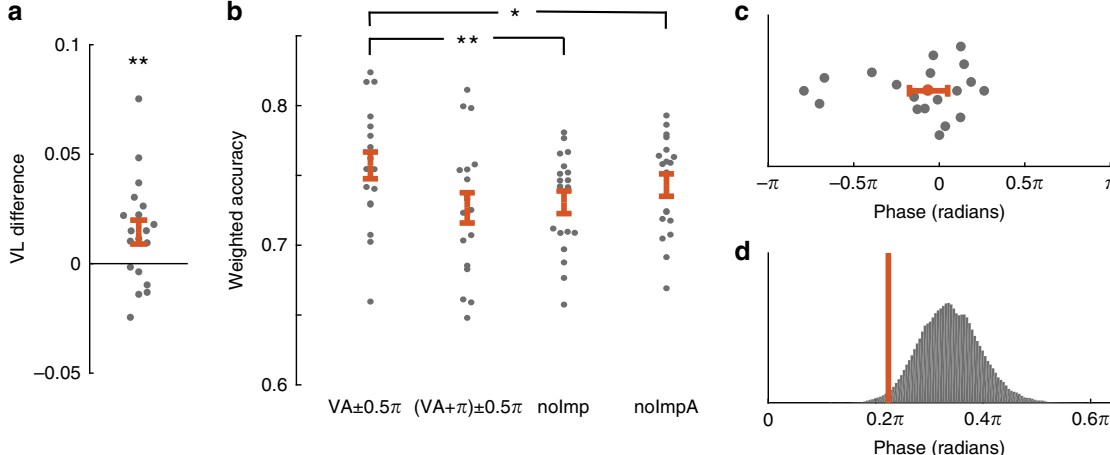

**Fig. 2 Behavioral results. a** The relatively change in vector length (VL) for accuracy. **b** Weighted accuracy for the VA, VA $+ \pi$, the no Impulse (noImp), a no Impulse after alignment to the VA (noImpA). **c** Linear phase plot of the phase differences between the vector phase of the accuracy and decoding. **d** Histogram of all permutations of absolute phase difference and observed phase difference (orange line). Error bars represent the standard error of the mean ($n = 19$; in (**c**) it represents the circular variance). **a** One-tailed paired t-test. **b** Two-tailed paired t-test, **p < 0.01, *p < 0.05.

following an impulse stimulus without considering oscillatory phase effects. Our re-analysis of the data set by Wolff et al.[23] provided additional support for the main findings in the present of our study (see Supplementary Figs. 7–10, Supplementary Notes 1 and 2, and Supplementary Discussion).

## Discussion

Theoretical proposals on the function of working memory have suggested that working memory content is represented in a distributed activity pattern across the brain, and that this representation is maintained in a cyclical manner[9]. We have performed a test of the phase-dependency of the representation of working memory content in human participants. To this end, we analyzed EEG data in which items had to be maintained in working memory during a retention period. Using classifier performance to probe and quantify working memory content representations in the brain, we found that the read-out of retained information by a memory-trace-boosting impulse stimulus was strongest for a limited range of ongoing oscillatory theta/alpha phases. Behavioral accuracies were also modulated by theta/alpha phase. Intriguingly, the phase at which decoding performance was highest closely matched the phase at which behavioral performance was at its optimum. The behavioral accuracy at the optimal phase ranges was also significantly higher compared to trials were the impulse stimulus was absent. These results suggest a phase-dependent representation of information content during working memory maintenance. Accordingly, strengthening the memorized information at the optimal phase of this cyclic representational mechanism systematically improved decoding of its neural representation and at the same time improved the accuracy of human memory performance. In addition, memory performance was better in trials with a memory-trace-boosting impulse stimulus at the right phase than in trials without.

Theta phase has been proposed to influence memory retention[9,13,25]. Specifically, it is suggested that neuronal activity at specific theta phases retains content information of the memorized item. In this way, relevant memory information can be grouped and separated from other irrelevant information[18,36,37]. Indeed, it has been shown that gamma power is strongest at specific phases[1,15,38,39], and that single neurons phase-lock to restricted theta phases during working memory paradigms[16,35]. While these studies show that neuronal activity centers around specific phases, the current study exploited a multivariate approach to demonstrate that it is the content of the information that is preferentially stored at specific phases within the oscillatory cycle (see also refs. [19,40]). The extraction of this phase-dependence in the post-impulse window could be a results of differences in the evoked response or a cross-talk between the impulse-evoked response and the ongoing brain activity that preserves the pre-stimulus phase (see also the Supplementary Discussion). The same research lines have suggested distinct roles for theta and alpha in working memory[9,25]. However, our study did not show this differentiation as the frequency bands involved covered both the alpha and theta range.

The improved decoding for restricted alpha/theta phases matched modulations of behavioral performance. Specifically, the phase of optimal behavioral performance mirrored the phase at which decoding performance was highest. Moreover, performance at these phases was higher compared to trials where no impulse stimulus was presented. This suggests that memory can be improved by activating the memory trace at the right moment in time. In line with this, brain stimulation studies suggest that stimulation effects depend on oscillatory phase[41,42], and that memory performance depends on the phase coherence between different regions[43]. Oscillatory phase may be the relevant parameter to consider for optimizing memory improvement interventions.

Previous studies have shown memory-related activity specific to restricted phases[16,19,39]. The present report adds two new insights to current knowledge. First, we demonstrated for the first time in humans that it is also the information content, and not only activity level, that is modulated by phase. This is compatible with a monkey neurophysiological study showing higher information content in spiking rates on distinct oscillatory phases in neurons in the PFC[18]. Second, we demonstrated the behavioral relevance of the phase-dependent coding of information content in working memory. Not only did we show a correlation between working memory performance and the low-frequency oscillation phase at which the impulse stimulus was shown, the simple intervention of presenting an impulse stimulus during the working memory delay also enhanced working memory compared to trials in which the impulse stimulus was absent. These findings provide crucial support for influential theories of working memory that more than two decades ago have proposed the phase-dependence of working memory information content and performance[9,44].

It has been suggested that multivariate classification of memorized information content after an impulse stimulus reflects the read-out of latent representations that are only present in the synaptic connectivity, without a level of ongoing neural activity[22,24,45]. The impulse stimulus allows for this read-out, similar to a sonar, but is hypothesized not to modulate the representation[45]. In our measurements, we show that the electrophysiological and behavioral effects of the impulse stimulus depend upon the phase of ongoing theta activity. This suggests, in line with a large body of evidence, that there is ongoing neural activity of which the amplitude in a subpopulation representing the memory trace is modulated by phase[38,46,47]. Considering this increased memory performance, the working memory representation was—at a minimum—maintained better following an impulse stimulus at the right phase than by not presenting the impulse stimulus (see also ref. [48]). It is difficult to imagine the impulse has a phase–dependent effect on decoding and working memory success, while at the same time the working memory neural trace would be latent and independent of phase. Of course it is possible that part of a neuronal representation is latent, but the part of the representation we probe in our study is most likely not.

Our findings are the first to show in human EEG a phase-dependent modulation of neural representation strength of memory content in the range of alpha and theta. We demonstrated that the cyclic representation of memory content has direct behavioral consequences for working memory performance. In addition, strengthening the memorized information at the optimal phase using an impulse stimulus not only systematically improved decoding of its neural representation, it also improved the accuracy of human memory performance beyond that observed without the impulse stimulus. Beyond their theoretical relevance, these novel mechanistic insights have direct implications for future therapeutic protocols aimed at improving human working memory.

## Methods

**Participants.** In total 20 participants completed the experiment (mean age: 24.4, range 18–45, 15 females). All had normal or corrected-to-normal vision. One participant was excluded due to low behavioral performance. All participants were informed about the study in advance and gave written informed consent prior to participating. The study was approved by the local ethical committee of the faculty of Psychology and Neuroscience at Maastricht University (ethical approval number: ECP-127 14_04_2013). Monetary compensation or participation credits were given to the participants for their time.

**Stimuli and procedure.** During the experiment participants sat in a Faraday shielded room at 60 cm distance from the monitor. Most procedures and stimuli were similar to Wolf et al.[23]. During the trial two or three different stimuli were

presented: a memory item, a target item, and potentially an impulse item. The memory item consisted of Gabor patches presented at 2.88 visual angle, with 0.62 cycles per degree at a random phase, at 20% contrast, presented for 200 ms. The orientation of the stimuli was randomly varied over all trials. The background was gray (RGB values: 127.5, 127.5, 127.5). The target item was identical to the memory item, except that the contrast was set at 100%. The target item had a specific angle offset from the memory item and could be at a −2, −4, −5, −7, −9, −12, −15, −20, −26, −34, 2, 4, 5, 7, 9, 12, 15, 20, 26, or 34 angle difference. The impulse stimulus consisted of a bullseye stimulus at a 100% contrast, at 0.62 cycles per degree, and was presented for 200 ms.

There were three different trial types: long impulse trials, short no-impulse trials and long no-impulse trials. During long impulse trials first the memory item was presented, the impulse stimulus was presented at a stimulus-onset asynchrony of 1300–1500 ms (at a uniform distribution). The target item was always presented at 2800 ms after the memory item onset. During the short and long no-impulse trials, no impulse stimuli were presented and the stimulus-onset asynchrony between the memory and target item was 1400 and 2800 ms respectively. The order of the trial types and angle differences were randomized. In total there were 1600 trials (800 long impulse trials, 400 short no-impulse trials, 400 long no-impulse trials). After the participants responded they heard a feedback sound monitoring their performance (880 Hz for correct and 440 Hz for incorrect, 50 ms duration). The next trial started after 1500 ms after the response of the participant. After every 24 trials the participant received feedback on their average performance of that block and could take a small break. After every five mini-blocks the participant had a longer break and allowed the experimenter to monitor the EEG signal.

**EEG acquisition and pre-processing**. Data was acquired with a 64-channel passive EEG system (Brain Products). Data was acquired at a 1000 Hz sampling rate with online band pass filter of 0.01–200 Hz (BrainVision Recorder software) using a BrainAmp Amplifier. Impedance was kept below 10 kiloOhm. The ground electrode was placed at Afz, and the online reference at the right mastoid. Three EOG were place at the outer canti of the eye and below the left eye to monitor eye movements. Pre-processing consisted of epoching the data between −3 and +2 around stimulus onset (for memory and impulse items). Then data was re-referenced to the average of all EEG channels, demeaned, resampled to 250 Hz, and bad trials were removed via visual inspection. Eye movements and muscle artefacts were corrected using ICA decomposition.

**MVPA analysis**. We trained a linear discriminant analysis (LDA) classifier using data at the time point of memory item presentation to discriminate between four 90 degree phase bin categories (using the COSMO toolbox[49]). This training was repeated twice for two different mean angle orientations per phase bin, similar as in (Wolff et al.[23]; categories for training 1:0–45, 45–90, 90–135, 135–180. Categories for training 2: −22.5–22.5, 22.5–67.5, 67.5–112.5, 112.5–157.5). We repeated the training for different time bins, ranging from 0 to 300 ms in steps of 4 ms (in each training three consecutive data points/12 ms were included in the analysis). Electrodes included in this analysis were P7, P5, P3, P1, Pz, P2, P4, P6, P8, PO7, PO3, POz, PO4, PO8, O2, O1, and Oz. These electrodes were used as we expected the strongest effects over occipital/parietal cortex in response to the visual impulse stimulus. Moreover, these channels were identical to the analysis of Wolff et al.[23]. We then tested whether the classifier could distinguish the orientation of the memory item based on the data presented after impulse onset matching the training and testing time. The results of this analysis is an accuracy for every single trial per time point (zero or one), which was used for the subsequent VL analysis[50]. Note that this memory-to-impulse cross-generalization did not result in significant orientation decoding in Wolff et al.[23]. However, we choose for this memory-to-impulse cross-generalization instead of any impulse-to-impulse decoding for two reasons: (1) our training data was independent of phase and solely contained the direct sensory response to the orientation stimulus. (2) We avoided splitting the data into phase bins which would highly reduce the power of our analysis. Moreover, we predicted modulations of orientation decoding and thus expected significant decoding only for a restricted phase range.

**VL analysis**. In the VL analysis we investigated whether the accuracy of decoding depends on the phase of ongoing oscillations at the time point of the impulse presentation. To extract ongoing oscillatory phase we performed a fast-Fourier transform (FFT) for frequencies ranging from 4 to 12 Hz in steps of 0.1 Hz with the Fieldtrip toolbox[51] using a Hanning taper. Data included three cycles of data prior to the onset of the impulse (thus the time window was frequency dependent). This entails repeating the FFT multiple times to ensure that for the estimated frequency stationary within the included time window would hold. We extracted the estimated phases for each trial which was correctly classified. Then we calculated the vector of all these phases. This vector has both a phase angle (VA) and a VL. The length varies between 0 (uniform distribution of phases) and 1 (all phases centered at one angle). Subsequently, we estimated for each data point what the expected random VL would be by shuffling the accuracy labels and calculating the VL for random phase-accuracy associations. We took as our random VL the median of 1000 repetitions. We calculated the difference of the VL with the random VL and performed a non-parametric Monte Carlo test. We corrected for multiple

comparisons using cluster thresholding as implemented in Fieldtrip (clusteralpha: 0.05; maxsum as dependent variable of the clustering).

**VA analysis**. We extracted mean accuracy for −0.5π to 0.5π around the VA and for −0.5π to 0.5π around the VA + π per participant at the center of the VL analysis effect (time/frequency point with significant cluster with the strongest effect). We compared the two phases bins in two ways. First, we determined an empirical mean accuracy per phase bin by calculating the mean accuracy per phase bin but centering determining the VA with the vector for each permutation (calculated before in the VL analysis). Each phase bin was then compared with its own empirical chance accuracy with paired samples t-tests. This analysis was performed as it is possible that even by picking the phase of a random vector higher mean accuracy for that phase bin are acquired. In a second permutation round, the testing labels in the decoding stage were randomized (n = 1000). The rest of the analysis was identical as for the original VA analysis (but using the original time/frequency point per participant).

**ERP analysis**. ERPs were calculated for the trials +/− 0.5π around the mean phase of the vector and around +/− 0.5π around the mean phase of the vector + π. These ERPs were calculated for both around the time of the Memory item and the Impulse item. We compared for each frequency and item type the mean phase and the mean phase + π with each other for the interval between 0 and 0.15 s averaging over the channels used for the decoding analysis using a non-parametric Monte Carlo simulation of the mean difference as dependent variable. We corrected for multiple comparison using cluster based thresholding (clusteralpha: 0.05; maxsum as dependent variable of the clustering).

**Behavioral VL analysis**. We extracted the accuracies. We only included trials with angle differences at which participants performed between 50 and 80% correct. The 50 and 80% thresholds were estimated by fitting a probit function to the data (using the modelfree toolbox[52]). The probit function was fitted on angle difference, using as dependent variable percentage leftward orientation responses (thus ranging from 0 to 100). Trials with angle differences between 20 and 80% leftward orientations were used in subsequent analysis (corresponding to accuracies between 50 and 80%). On average 38.0% of the trials were removed (range: 25.3–55.4%). We repeated the VL analysis for the accuracy using the frequencies with the most significant effects but by using the angle differences as a weighting factor (so that phases of more difficult trials had a higher influence). The factor was calculated by normalizing the angle differences between the memory item and the probe between 0 and 1 (using the mat2gray function in matlab).

**VA comparison**. We calculated the difference between the VAs of the MVPA and the behavioral analysis. A V-test was performed to identify whether the phase differences were uniformly distributed on zero. This test is similar to the Rayleigh test, but preferred when there is an expected phase direction. Moreover, the test will be non-significant for data that is either not uniform or has a different mean phase. Additionally, we tested via permutation tests whether the phase difference was non-uniform around zero using the V-statistics as dependent variable (using the VA of the permutation calculated in both VL analyses).

**Behavioral VA analysis**. Firstly, we repeated the VA analysis based on the decoding for the behavior. Here, the dependent variable was the weighted accuracy calculated by taking the sum of the weights for the accurate trials and dividing by sum of all the weights. The permutation tests were based on random shuffling the phase labels, recalculating the aligned weighted accuracies. In a second step we were interested whether the impulse stimulus at the VA improved behavioral performance relative to having no impulse stimulus at all (i.e., the long no-impulse trials). Therefore, we recalculated the weighted accuracies including all trials (as we wanted to directly compare accuracies over different trial types, we could not restrict the analysis to a relevant optimized range). As the long no-impulse trials had half the trials, the VA analysis was repeated 1000 times using the same amount of trials as present in the no-impulse trials. The average weighted accuracies were used. Then a repeated measures ANOVA was used including the factor item condition (levels: VA, VA + π, no-impulse) was performed. Pairwise comparisons followed this analysis, using Bonferonni correction. Finally, the original behavioral VA condition was compared to the median of 1000 arbitrarily calculated weighted accuracies of the VA of the no-impulse trials (using the phases of the impulse trials).

**Reporting summary**. Further information on research design is available in the Nature Research Reporting Summary linked to this article.

## Data availability
The EEG data and behavioural logfiles related to the figures are available in the repository of Maastricht University (https://hdl.handle.net/10411/NUNJML).

## Code availability
The code related to the figures are available in the repository of Maastricht University (https://hdl.handle.net/10411/NUNJML).

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

## Acknowledgements

This work was supported by The Netherlands Organization for Scientific Research (NWO; 453-15-008). We would like to thank Mark Stokes and Michael Wolff for sharing their data, providing valuable feedback and contributing to the manuscript.

## Author contributions

Conceptualization: S.t.O.; Methodology: S.t.O.; Formal analysis: S.t.O.; Writing—Original draft: S.t.O.; Writing—Review and Editing: P.d.W. and A.T.S.; Supervision: P.d.W. and A.T.S.; Funding Acquisition: A.T.S.

## Competing interests

The authors declare no competing interests.
