## [Peer Review File · Nature Communications]

Reviewers' Comments:

Reviewer #1:

Remarks to the Author:

"Phase-dependent enhancement of working memory content and performance," Sanne ten Oever De Weerd, and Sack. This manuscript describes a study in which subjects performed visual working memory for oriented grating stimuli while the EEG was recorded and single pulses of TMS delivered during the delay period. On trials whose sample-to-test orientation difference yielded performance of < 80% correct decoding was superior in a broad band from ~7-14 Hz for a 10 msec-long epoch at 130 msec after TMS and from 4-10 Hz for a 30 msec-long epoch at 190 msec. There are some conceptual and analytic/empirical concerns that dampen this reviewer's enthusiasm for this work. Most prominent is the assumption that underlies the interpretation of the results, which is that "information content (of working memory representations), not only activity level, ... is modulated by phase." This would seem to be a conflation of decoder performance and "information content." The first-order logical problem is evident in the fact that this superior decoding is only observed during these brief windows and, importantly, not at the time of the behavioral response. If instantaneous decoding performance related to "information content" in some absolute sense then these "blips" of elevated information content would not influence performance, for the simple reason that they "go away" long before the test stimulus. If one were to argue that the putative TMS-evoked increase in information content is preserved for the remainder of the trial in a format that is invisible to these methods, a logical corollary of this argument would then have to be that the "blips," themselves, don't correspond to the information, per se, but rather to a momentary state of being more amenable to read-out. But this, of course, is the same as saying that the TMS influences the "activity level" of the representation, and that its level of information is latent throughout the trial. It's also worth noting that the 'boost of information' interpretation is at odds with the interpretation given to their similar results by Wolff and colleagues – they very explicitly use the sonar metaphor to argue that their flash reveals the momentary state of the otherwise latent representation, but without altering it. (The accompanying commentary by Serences and Rademaker also makes this point explicitly.) There are two additional conceptual points that would need to be addressed. Sprague and Serences have argued in a recent paper that information theory argues against the ability of a system to internally generate more information from a low-information representation, unless it has access to additional information from another source. Second, there is a growing body of literature suggesting that it may be confidence, rather than stimulus information, that varies with alpha phase.

The last point facilitates a transition into analytic questions: to address the question of discriminability vs. confidence, a single-detection analysis of the behavioral data might be needed.

The effects, both decoding and behavioral, are quite modest, and rendered all the more equivocal by the exclusion of > 80% accuracy trials. What proportion of trials was excluded?

Perhaps most problematic for the interpretation of the results is, per Figure 1.B, that classification of VA+pi is the same percentage below chance as VA is above chance. This suggests that it may merely be the case that excitability of the neural ensemble encoding the representation oscillates at the frequency(s) in question. This is reminiscent of what Grill-Spector has observed for various stimulus-category representations with fMRI.

It'd be helpful to know if, on a subject-by-subject basis, an individual's phase angle of maximal decoding corresponded to her/his phase angle of maximal performance on no-TMS trials. (For this, one would need to calculate instantaneous phase at the time of test onset.) Indeed, if trials were broken out by performance AND by TMS status, and timelocked to test onset, would other oscillatory structure emerge, and would the patterns look different between TMS and no-TMS?

It's not the case that "the behavioral consequences of this amplification were left unstudied by ref. 25. In 25, the representation of interest was the one of two that was NOT relevant for a memory probe, and TMS produced an elevated false-alarm rate on trials when this item was presented as a lure.

"using a FFT approach" is simply too vague an explanation for how instantaneous phase angle was calculated.

Figure 2. seems to be missing panel D.

Reviewer #2:

Remarks to the Author:

This manuscript addresses the neurophysiological basis of human working memory by using EEG recordings, decoding analyses, and an innovative "impulse stimulus" paradigm described in earlier papers.

The authors argue that the results show that (i) "neural memory content representation fluctuates in a phase-dependent manner at theta/alpha frequencies" and that (ii) "its neural representation as well as the corresponding working memory performance can be enhanced by a phase-specific sensory impulse stimulus".

These may be significant advances in uncovering working memory mechanisms widely interesting for general audience. I would, however, tentatively argue that the evidence for (i) is questionable in the present version of the manuscript and that for (ii) there is room for being more conclusive/reliable. I outline below the concerns for these arguments.

1. Reliability of the observations: The 'primary outcome' inferential statistic is the data in Fig. 1A, where significance is established with cluster statistics. While cluster-based permutation statistics account for multiple comparisons, they "do not establish significance of effect latency or location" (<https://www.ncbi.nlm.nih.gov/pubmed/30657176>). This should be carefully considered in the inferences. If I understand correctly, the data used in Fig. 1B, and suppl. figs. 3 and 5, are selected for the clusters in Fig. 1A and thus these analyses have no inferential value per se but rather act as post hoc statistics/illustrations for the main effect.

1b. Observing significant effects in such narrow time windows raises a question of their reliability, which has not been estimated here. It would be important to corroborate the findings both with a replication cohort and appropriate reliability analyses, and by using another statistical approach - preferably one that enables inferences about the frequencies and latencies of the effects. In the current replication based on earlier data, the authors found the effects in completely different latency range and these data failed to reproduce the behavioral performance finding.

2. Evidence for a relationship with oscillations: the frequency window for the analyses could be expanded to illustrate more convincingly that the effects really are band-limited and attributable to an oscillatory rather than broadband process. The first one of the main effects seems to span frequencies from theta to low-beta bands.

2b. I am curious about the relationship of these findings with the responses evoked by the impulse stimulus. Do the alpha and theta frequency components here relate to the waveforms of the early and late, respectively, peaks of the evoked response? What is the amount of information about the sample

stimulus that is decodable phase-dependently at the latencies of the impulse stimulus in trials where the impulse is absent? Does the phase dependence of the impulse-stimulus decodability reflect (i) differences in the actual evoked responses or (ii) a cross-talk between the impulse-evoked response and ongoing brain activities that preserve the pre-stimulus phase and become superimposed with the evoked response.

3. It would be important to address the anatomical localization of these effects, preferably with source localization but at least with scalp topographies. The authors acquired 64-channel EEG but used only 17 parieto-occipital channels for decoding - why is this? What was the rationale (a priori vs. trying many combinations) for the channel selection?

3b. In relation to Suppl. Fig. 4, the authors claim "Instead, the distributed response profile evoked by the impulse more closely matches the response profile of the memorized item when the impulse is presented within a specific restricted phase range.": this is something that could be explicitly tested by assessing whether the similarity of the sample- and impulse-responses is phase dependent.

3c. The conclusion "Our findings are the first to show in human parietal-occipital cortex a phase-dependent modulation of neural representation strength of memory content in the range of alpha and theta." is not only questionable with respect to frequency localization (see 1. and 2.) but also unwarranted in terms of the anatomical localization: there is no evidence that the signals yielding the decoded information were produced in the parieto-occipital cortex. These signals could be produced almost anywhere in the brain and be picked up in the authors channel-selection via volume conduction. Addressing the decodability across the entire 64-channel data and preferably with source reconstruction methods is fundamental for drawing any anatomical inferences.

4. Overall the data are rather scarcely illustrated. Instead of bar plots, there are more modern ways to illustrate the distributedness of the findings. For the behavioral results, it would be more tangible to express them in terms of hit rates and the phase dependence of the hit rates per se. The circular phase plots (Fig. 2C, Suppl. Fig. 7B) would in my opinion be better shown on a linear phase axis and by illustrating the reliability by confidence limits (surrogate data and/or bootstraps).

Reviewer #3:

Remarks to the Author:

The present manuscript, titled "Phase-dependent enhancement of working memory content and performance," follows on previous work showing (1) that the content of working memory can be decoded from EEG data based on the response to high-intensity impulse stimulus presented during the delay between the sample item and the target item (i.e., while the sample is being held in working memory) and (2) that working memory is linked to low-frequency oscillations. Their results suggest that the distributed neural activity associated with memorized information waxes and wanes with the phase of these low-frequency oscillations. That is, classification of the sample item when an impulse stimulus was presented during the memory delay (based on a linear classifier) was more accurate at specific phases of neural activity in the theta/alpha range. Their results also suggest that behavioral performance (i.e., behavioral accuracy) was better on trials when the impulse was presented during phases associated with better decoder accuracy. Their results thus indicate that impulse stimulation has the potential to improve working memory, if the timing of the presentation is properly aligned with underlying oscillatory activity. These are certainly exciting conclusions, and the authors took the commendable step of replicating their results in a second dataset. However, there are several issues/concerns that I would like the authors to comment/address before I can be convinced of their results.

1. In the introduction (page 3): "...the memorized information can be decoded from EEG electrodes over occipital and parietal lobes." It would be helpful to add a sentence or two here about why these might have more information than frontal electrodes. Also, the authors might consider removing the location of the electrodes from the abstract, as there's no explanation there for why they restricted recordings to occipital and parietal regions.
2. In the introduction (page 3): "In addition, given the role of low frequency oscillations in working memory and existing theoretical proposals..." This is vague. Can you be more specific regarding these theoretical proposals?
3. In the results (page 5): "...we first estimated phase at impulse onset... using an FFT approach including three cycles of data prior to impulse onset." Please clarify. For each frequency, you used data equivalent to three cycles, meaning the amount of data for each frequency was different? Also, an FFT provides the frequency content of a signal across multiple frequencies, unlike, e.g., a wavelet, which is centered on a specific frequency. Why were multiple FFTs run here?
4. Figure 1A. Here, the authors present the phase-specific results (i.e., whether there was phase consistency prior to the impulse stimulus on correctly classified trials). It was previously stated that the classifier could identify the orientation of the working memory item in 12 ms data bins collected 0-150 ms following the impulse stimulus. Here, the figure shows 0-250 ms, with the significant VL for time points near or after the offset of the impulse stimulus. What was the corresponding, non-phase dependent classifier accuracy at each time point? Second, is it possible to get corresponding VLs for the incorrectly classified trials, and what is the difference in VA between correctly and incorrectly classified trials? Finally can the authors speculate at all regarding the timing of these effects relative to phase measurements prior to impulse stimulation (i.e., 150 ms earlier)? The results presented in Supplemental Figures 6 and 7, show significant results in the 150ms period corresponding with impulse stimulation.
5. Figure 1B. What frequency and time point are being represented here?
6. Are there differences in oscillatory power just prior to impulse stimulation that might also influence classifier accuracy?
7. Perhaps the biggest weakness reported in the results is that the phase of oscillatory activity associated with better classifier and behavioral accuracy differs across participants. The authors speculate that this could be related to methodological disadvantages of EEG, such as volume conduction and the superposition of electric potentials, but I don't find this explanation satisfying. Are the authors suggesting that they are detecting different memory-related signals across participants that are all predictive of classifier and behavioral accuracy? If these effects trace back to a single neural basis, then how would volume conduction or the superposition of electrode potentials lead to subject-specific phases? Although I'm not satisfied with the explanation, my concerns are somewhat mitigated by the subject level convergence for the phases related to better classifier and behavioral accuracy.
8. In the results (page 7): "We tested the observed classifier accuracy to the median of the chance distributions of accuracies obtained after 1000 permutations of the correct/incorrect classification labels at the stage of calculating the VL." To test whether the observed VL is higher than that expected by chance, shouldn't you see if it exceeds 950 (for $p < 0.05$) of the values in the test distribution (i.e., the randomly permuted distribution)? Instead the authors use the median from this permutation approach as a benchmark for further statistical testing. Please explain. These same

questions/concerns apply to the analysis that examines pre-classifier phase and behavioral accuracy.

9. Figure 2. The figure doesn't match the caption. The caption refers to (D), which seems to be (B), and there is no (D).

10. In the methods (page 13), there's a typo: "long impulse trials, short no-impulse trials, long impulse trials." Should be: long impulse trials, short no-impulse trials, long no-impulse trials.

11. In discussion (page 11): "Our results are also of direct relevance for the discussion where in the brain working memory content is maintained..." I don't think this (and the paragraph that follows) is a fair statement, since it seems the authors only examined electrodes over occipital and parietal regions.

We thank the reviewers and editor for their support and suggestions. We addressed all concerns of the reviewers in our updated manuscript (highlighted in yellow). Please find detailed explanations below.

Reviewer #1 (Remarks to the Author):

“Phase-dependent enhancement of working memory content and performance,” Sanne ten Oever De Weerd, and Sack. This manuscript describes a study in which subjects performed visual working memory for oriented grating stimuli while the EEG was recorded and single pulses of TMS delivered during the delay period. On trials whose sample-to-test orientation difference yielded performance of < 80% correct decoding was superior in a broad band from ~7-14 Hz for a 10 msec-long epoch at 130 msec after TMS and from 4-10 Hz for a 30 msec-long epoch at 190 msec. There are some conceptual and analytic/empirical concerns that dampen this reviewer’s enthusiasm for this work. Most prominent is the assumption that underlies the interpretation of the results, which is that “information content (of working memory representations), not only activity level, ... is modulated by phase.” This would seem to be a conflation of decoder performance and “information content.” The first-order logical problem is evident in the fact that this superior decoding is only observed during these brief windows and, importantly, not at the time of the behavioral response. If instantaneous decoding performance related to “information content” in some absolute sense then these “blips” of elevated information content would not influence performance, for the simple reason that they “go away” long before the test stimulus. If one were to argue that the putative TMS-evoked increase in information content is preserved for the remainder of the trial in a format that is invisible to these methods, a logical corollary of this argument would then have to be that the “blips,” themselves, don’t correspond to the information, per se, but rather to a momentary state of being more amenable to read-out. But this, of course, is the same as saying that the TMS influences the “activity level” of the representation, and that its level of information is latent throughout the trial. It’s also worth noting that the ‘boost of information’ interpretation is at odds with the interpretation given to their similar results by Wolff and colleagues – they very explicitly use the sonar metaphor to argue that their flash reveals the momentary state of the otherwise latent representation, but without altering it. (The accompanying commentary by Serences and Rademaker also makes this point explicitly.) There are two additional conceptual points that would need to be addressed. Sprague and Serences have argued in a recent paper that information theory argues against the ability of a system to internally generate more information from a low-information representation, unless it has access to additional information from another source. Second, there is a growing body of literature suggesting that it may be confidence, rather than stimulus information, that varies with alpha phase.

We thank the reviewer for his/her clear argumentation. While we agree with a number of points raised by the reviewer, we also respectfully disagree with some others. We will address each of these points in our detailed response below. But first, we would like to point out that the impulse stimulus used in our study was not a TMS pulse, but a visual stimulus. As in our previous manuscript, we defined the impulse stimulus as a visual bull’s eye stimulus (see lines 64-66), and the term TMS was never used in

our manuscript to describe the impulse stimulus. To avoid that also other readers mistake our impulse for a TMS intervention, we now made sure to state clearly that we use a visual impulse stimulus (see e.g. line 19).

Having said this, we now would like to address the points related to the debate whether our results can be interpreted in terms of activity level or information content. The reviewer argued that:

- 1) *‘If instantaneous decoding performance related to “information content” in some absolute sense then these “blips” of elevated information content would not influence performance, for the simple reason that they “go away” long before the test stimulus.’*

We understand the reasoning, however, our results clearly show that the impulse stimulus did have an influence on behavioral performance. After presenting the impulse stimulus at the phase where decoding was optimal, the behavioral performance was higher with as compared to without the impulse stimulus. This shows that when directly comparing to not having an impulse stimulus, the impulse had an influence on the memorized information.

- 2) *‘...the “blips,” themselves, don’t correspond to the information, per se, but rather to a momentary state of being more amenable to read-out.’*

We would like to argue that even if we only had shown that the impulse stimulus just allowed a brief peak into the nature of the WM representation (by a brief opportunity for enhanced read-out), our findings would have been quite significant. In fact, we show for the first time that discriminability of information in WM oscillates. As communication across networks and areas occurs at a small time scale, this means that WM information is accessible only at specific moments in time, to which the other areas need to be tuned. In addition to that, we found that the presence of the impulse stimulus in fact did have behavioral consequences, as explained in response to the previous point.

We agree however that our terminology of ‘boosting information’ was ambiguous. In line with the argumentation of Sprague and Serences we can indeed not conclude that we necessarily increase the amount of information in the system during the course of the trial. However, we did find an improved behavioral performance compared to the absence of a stimulus. Considering this enhanced performance, the working memory representation was - at a minimum - maintained better following an impulse stimulus at the right phase as compared to not presenting the impulse stimulus. Hence, to some extent, the impulse stimulus counters decay during the WM interval.

In sum, we consider the insights from our experiment on WM representations, neural communication during WM maintenance, and the effect of the impulse stimulus on WM maintenance highly relevant. However, the terminology ‘boost of information’, without clarification, might obscure these important points. We therefore changed the term boost when appropriate (see the change for example in the title, abstract end of the introduction (page 3)).

- 3) *'But this, of course, is the same as saying that the TMS influences the "activity level" of the representation...'*

We agree with the reviewer, as the impulse stimulus was indeed meant to influence activity levels of the representation, which contain information about the WM item. We like to make some clarifications regarding this point.

We acknowledge that the way we phrased a contrast between activity levels and information, may not have been clear. The point we wished to make is that many previous studies have considered activity levels in a univariate way, by considering only single neurons, or by averaging over (e.g., EEG) electrodes or fMRI voxels. Any high impulse stimulus (TMS or visual) would result in an increase in overall univariate activity levels measured by a visible ERP. Univariate analysis however does not give access to the decoding of information. Multivariate pattern analysis instead has been able to decode stimulus information in a variety of sensory/perceptual experiments. Indeed, overall univariate analysis does not give us any significant effect (no ERP phase differences), but multivariate analysis does. Our data indicate that there are no net univariate activity increases, but instead representation-specific activity increases, as demonstrated by our classification analysis. Thus, what increases the discriminability of the distributed information at subsets of phases is embedded in an unchanged averaged level of ongoing WM related activity. Hence, our study shows for the first time that WM content can not only be read-out, but that the read-out efficiency is theta-phase dependent, and that the impulse stimulus that enables read-out also enhances WM performance when presented at the optimal theta phase. As stated before, these findings are informative both with respect to the properties of the WM representation, and the constraints imposed on neural activity that is involved in WM maintenance. We clarify the above points in the paper on lines 68-74.

- 4) *'..., and that its level of information is latent throughout the trial.'*

We understand 'latent' here as a memory trace that is represented in a pattern of synaptic connectivity without a level of ongoing neural activity. In our measurements, we show that the electrophysiological and behavioral effects of the impulse stimulus depend upon the phase of ongoing theta activity. This suggests, in line with a large body of evidence, that there is ongoing neural activity of which the amplitude in each channel/subpopulation/neuron is modulated by phase. It is difficult to imagine the impulse has a phase-dependent effect on decoding and WM success, while at the same time the WM neural trace would be latent and independent of phase. We now have discussed these important points on page 12, lines 286-299.

- 5) *'information theory argues against the ability of a system to internally generate more information from a low-information representation'. '...it may be confidence, rather than stimulus information, that varies with alpha phase.'*

We agree that no new information is added due to the impulse stimulus. As elaborated above in points 2 and 3, we believe the impulse stimulus maintains the amplitude of the WM

informative signal in all electrodes compared to the noise (perhaps with a gain factor), and in this way enables the pattern of signals to be read out more efficiently. We have included these ideas in our Discussion (page 12).

The last point facilitates a transition into analytic questions: to address the question of discriminability vs. confidence, a signal-detection analysis of the behavioral data might be needed.

It is possible that enhanced signal precision/reliability also leads to the subjective experience of more confidence. However, our two alternative forced choice experiment did not include a measure of confidence level. Therefore, it is not possible to address this topic and goes beyond the scope of the present paper.

The effects, both decoding and behavioral, are quite modest, and rendered all the more equivocal by the exclusion of > 80% accuracy trials. What proportion of trials was excluded?

A mean of 38.0% (standard deviation of 9.2%; range 25.3% - 55.4%) of trials were removed. This information has been added to the manuscript (page 16, line 422). Note that this exclusion was only for the behavioral analysis relating to the vector length, not for the decoding. As explained in our paper (lines 181-184), *we selected trials with orientation differences between the sample and test item corresponding to performance levels below 80%. This selection of more difficult trials was necessary because more easy trials (>80% correct performance) in which performance would already be near ceiling preclude the detection of phase dependent performance benefits.* We suggest this is a valid argument for selecting trials.

Perhaps most problematic for the interpretation of the results is, per Figure 1.B, that classification of VA+pi is the same percentage below chance as VA is above chance. This suggests that it may merely be the case that excitability of the neural ensemble encoding the representation oscillates at the frequency(s) in question. This is reminiscent of what Grill-Spector has observed for various stimulus-category representations with fMRI.

It is well known that measured oscillations likely represent oscillations in excitability in the generative neural population. Indeed, it is possible that at higher excitabilities it is easier to activate the representation containing the information of the memorized stimulus type. In our study however, we found that it is the distributed representation itself that oscillates and *not* the averaged activity in the overall region representing the WM item, as the overall ERPs between the optimal and non-optimal phase did not differ. Therefore, our findings go far beyond the mere link between excitability fluctuations and oscillations. Our data indicate that distributed patterns of activity that represent WM information are maintained by an active, oscillating mechanism, with the consequence that the access to the information itself (or the possibility for successful readout) fluctuates over time. Hence, rather than overall changes in activity levels, we show that the strength of the WM information itself – or the

accessibility for readout – is modulated. This has so far only been hypothesized in influential theories (e.g., Ole Jensen and Lisman, *Neuron*, 2013) but never been shown empirically. Our demonstration of the oscillation in accessibility of the WM information is therefore theoretically relevant, and has important implications for the way in which networks have to communicate during WM maintenance.

It'd be helpful to know if, on a subject-by-subject basis, an individual's phase angle of maximal decoding corresponded to her/his phase angle of maximal performance on no-TMS trials. (For this, one would need to calculate instantaneous phase at the time of test onset.) Indeed, if trials were broken out by performance AND by TMS status, and timelocked to test onset, would other oscillatory structure emerge, and would the patterns look different between TMS and no-TMS?

We ran the analysis suggested by the reviewer. We extracted the phase angles at test onset for the no-Impulse trials for the frequencies of the maximal decoding and investigated if behavioral performance was modulated. As in the behavioral analysis of the paper we calculated the vector angle weighted by the accuracy. For this analysis we did not find a significant effect ($t(18) = -1.23$, $p = 0.882$). This is not unexpected if one accepts that the sensory representation in parieto-occipital cortex following the initial encoding of the stimulus differs from the mnemonic parieto-occipital representation of the same stimulus.

It's not the case that “the behavioral consequences of this amplification were left unstudied by ref. 25. In 25, the representation of interest was the one of two that was NOT relevant for a memory probe, and TMS produced an elevated false-alarm rate on trials when this item was presented as a lure.

We regret being unspecific on the reference and adjusting it accordingly (page 3).

“using a FFT approach” is simply too vague an explanation for how instantaneous phase angle was calculated.

We made the explanation clearer in the manuscript (page 5, line 96-98). We now elaborated in the paper that we “*estimated phase at impulse onset (for frequencies ranging from 4-15 Hz) by extracting the phase from the Fast-Fourier Transform of data in a three-cycle time window preceding impulse onset*”. Please also find the detailed description in the methods (page 15).

Figure 2. seems to be missing panel D.

We thank the reviewer for noticing and adjusted it accordingly.

Reviewer #2 (Remarks to the Author):

This manuscript addresses the neurophysiological basis of human working memory by using EEG recordings, decoding analyses, and an innovative "impulse stimulus" paradigm described in earlier papers.

The authors argue that the results show that (i) "neural memory content representation fluctuates in a phase-dependent manner at theta/alpha frequencies" and that (ii) "its neural representation as well as the corresponding working memory performance can be enhanced by a phase-specific sensory impulse stimulus".

These may be significant advances in uncovering working memory mechanisms widely interesting for general audience. I would, however, tentatively argue that the evidence for (i) is questionable in the present version of the manuscript and that for (ii) there is room for being more conclusive/reliable. I outline below the concerns for these arguments.

1. Reliability of the observations: The 'primary outcome' inferential statistic is the data in Fig. 1A, where significance is established with cluster statistics. While cluster-based permutation statistics account for multiple comparisons, they "do not establish significance of effect latency or location" (<https://www.ncbi.nlm.nih.gov/pubmed/30657176>). This should be carefully considered in the inferences. If I understand correctly, the data used in Fig. 1B, and suppl. figs. 3 and 5, are selected for the clusters in Fig. 1A and thus these analyses have no inferential value per se but rather act as post hoc statistics/illustrations for the main effect.

We fully agree and this is also why we restricted our analysis to the frequency ranges of interest (theta/alpha). Location was never a factor in the clustering at the level of the VL analysis as we used the occipital/parietal channels as the features in our decoding approach (repeating the decoding for different time points), leaving only a single time course of decoding performance per participant. We also did not intend to make strong claims about latency differences based on the clustering results. Our primary result is that the decoding is phase dependent. We regret if in the conclusion or anywhere else in the paper, we were too strong in drawing conclusions related to locations and latencies. Throughout the manuscript these conclusions were adjusted accordingly and we now aimed to specifically state the limitations of this clustering approach (see page 5, lines 112-114). Indeed, Figure 1B, supplementary figure 3 and 5 are post-hoc statistics for the main effect, and we have now stated this explicitly (see e.g. line 147 and the figure legend).

1b. Observing significant effects in such narrow time windows raises a question of their reliability, which has not been estimated here. It would be important to corroborate the findings both with a replication cohort and appropriate reliability analyses, and by using another statistical approach - preferably one that enables inferences about the frequencies and latencies of the effects. In the current

replication based on earlier data, the authors found the effects in completely different latency range and these data failed to reproduce the behavioral performance finding.

We appreciate these points made by the reviewer and fully agree. Therefore, the reliability question is addressed to a considerable extent in the current manuscript. Specifically, we compared the findings with another similar dataset (supplementary materials). We believe that our main findings in our new dataset are largely in line with our re-analysis of the older dataset by Wolff et al, using the same analysis approach (as described in our current paper) for the two datasets. Specifically, we found in both datasets that theta/alpha phase modulated classification performance. Moreover, these frequency ranges also influenced behavioral performance. In the dataset of Wolff et al. the behavioral analysis showed a p-value of 0.07 and in our own data set we found a significant effect with $p = 0.008$. While the outcome for Wolff is not significant, considering the arbitrariness of the p-value cutoff (Goodman, 1999; Bakan, 1966) and the additional statistical support from the current cohort it is difficult to maintain the null hypothesis and we feel that both data sets strengthen each other's behavioral findings.

One analysis outcome that differed between studies relates to the time windows within which classification success occurred. It is not entirely clear why in the dataset of Wolff et al., the time window permitting phase-dependent decoding occurs earlier after the impulse stimulus. This could be related to differences in exact paradigm design. For example, in our paradigm, the moment of impulse stimulus presentation was less predictable than in Wolff's experiment (uniformly jittered respectively between 1100-1300ms versus 1170 or 1230ms). This may have led to predictabilities in stimulus onset of the impulse stimulus in Wolff et al., leading to earlier responses to the impulse stimulus (Lange, 2009; Miniussi et al., 1999), and to earlier decoding success. This explanation is supported by the ERPs in both datasets: the dataset of Wolff has earlier ERPs in response to the impulse (see figure below).

Nevertheless, as stated above, the findings from our analysis as applied to Wolff's data and our own are highly compatible. Please also note that the windows of enhanced decoding are within a time range following the impulse stimulus within which sensory effects of the impulse stimulus would have their maximal effect. Hence the effects in both studies occur in acceptable time windows following the impulse stimulus. These considerations have been included in Supplementary Discussion.

To further address the reliability of the effect in our own data, we split the data in odd and even trials and investigated the reliability within and outside the significant clusters (average Spearman correlation over participants for data points within and outside the significant clusters). We found a split-half reliability of 0.168 within the cluster, while outside the cluster range, the reliability was 0.004. This reliability analysis was added to the manuscript (page 5, lines 117-120).

We agree that the mass cluster approach does not permit clear-cut conclusions about the delay before onset of enhanced classification (latency), and about the frequency ranges within which decoding is enhanced (page 5, line 112-114). The reviewer suggests to do a statistical analysis in which conclusions can be drawn on the latency and frequency level. While this is an interesting suggestion, originally we were not interested in latency differences or reaching conclusions about details of the frequency content (see answer to the first point). Instead, we opted for the most powerful analysis approach which is the mass cluster analysis, which is blind to frequency and latency.

To still offer a statistical approach that provides information about both the latency and frequency, we used the threshold-free cluster enhancement (TFCE) method for clustering (Smith & Nichols, 2009; Mensen & Khatami, 2012). In this method, a maximum criterion for multiple comparisons is applied, but only after filtering the data, such that the clusters either with high statistical values or with a big cluster extent are emphasized. To apply this method, one has to determine two parameters that can be referred to as cluster height (H) and cluster extent (E), which when set optimally minimize the False Alarm Rate and maximize the Hits. In order to set these parameters independently of our own data, we used the expected variations of Wolff's data by taking the data at the stage of the VL calculation for decoding with randomized phase labels (i.e. empirical chance VL data). We added two Gaussian effect clusters of similar size as the original clusters and random noise within mean and standard deviation of the original data. Then we calculated with the TFCE the p-values and repeated this for 100 times. We calculated the optimal H and E, thus minimizing the False Alarm rate and maximizing the Hits of the effect (H and E both varied between 0.4 and 2 in steps of 0.1). See the result of this parameter search below:

Based on this search we determined that an H value of 1.8 and an E value of 0.8 maximizes hits, and minimizes false alarms (similar to empirical findings of previous papers (Mensen & Khatami, 2012)). Using these H and E values for the TFCE clustering analysis of our own data, we found clusters in similar time windows and frequency ranges as in our original cluster mass method (as shown below):

However, as the TFCE clustering analysis is a bit less powerful than the cluster mass method, we merely reached a trend (lowest corrected p-value = 0.065). This means that also this analysis cannot be used to derive from it definitive statements about location and latencies. Nevertheless, we feel that the outcome of this analysis supports the time windows revealed by our original analysis, and the general finding of decodability of the WM trace within these time windows following the impulse stimulus. If the reviewers and editors find it useful to include the trends from TFCE analysis in the Supplementary Materials, we are happy to do so.

In summary, to address the issue of reliability in the paper, we have added our reliability analysis to the Results, and have added a paragraph in the supplementary discussion where we highlight the replicability of main findings across Wolff's data set and our own, as well as the results of the reliability analysis based on a 50/50 split of our own dataset. We also acknowledge differences between our data and those of Wolff et al. in terms of the onset of the decodability time window following the impulse

stimulus, included a discussion of factors that can explain these differences, and in Results took into account the fact that latency differences among clusters or brain location differences cannot be addressed with our analysis.

2. Evidence for a relationship with oscillations: the frequency window for the analyses could be expanded to illustrate more convincingly that the effects really are band-limited and attributable to an oscillatory rather than broadband process. The first one of the main effects seems to span frequencies from theta to low-beta bands.

We repeated the analysis for both the beta (15-25) and the gamma (25-45) range. For both ranges no significant effect was found (beta lowest cluster value: 0.318, $p = 0.166$, gamma lowest cluster value: 0.144, $p = 0.411$). We added the figure for the longer band to the supplementary materials (Supplementary Figure 4C+D).

2b. I am curious about the relationship of these findings with the responses evoked by the impulse stimulus. Do the alpha and theta frequency components here relate to the waveforms of the early and late, respectively, peaks of the evoked response? What is the amount of information about the sample stimulus that is decodable phase-dependently at the latencies of the impulse stimulus in trials where the impulse is absent? Does the phase dependence of the impulse-stimulus decodability reflect (i) differences in the actual evoked responses or (ii) a cross-talk between the impulse-evoked response and ongoing brain activities that preserve the pre-stimulus phase and become superimposed with the evoked response.

These are some exciting questions, which we try to address below.

‘Do the alpha and theta frequency components here relate to the waveforms of the early and late, respectively, peaks of the evoked response?’

Looking at supplementary figure 5A (lower panel, impulse item), it is indeed interesting to note that the early window first time window of decodability in the alpha range was centered at 0.14s, which roughly matches the first positive peak in the ERPs at 0.12s. Likewise, the later time window of decodability was centered on 0.2s, but was of a broader range so overlapped with the first negative peak of the ERP (centered at 0.18ms).

‘What is the amount of information about the sample stimulus that is decodable phase-dependently at the latencies of the impulse stimulus in trials where the impulse is absent?’

It is difficult to extract the phase at the latency of the impulse stimulus as the impulse stimulus onset time was jittered for over 200 ms. So this would entail a full theta cycle of which the impulse stimulus is expected. So we cannot reasonably extract a phase-dependent coding at the time the impulse would have occurred.

Does the phase dependence of the impulse-stimulus decodability reflect (i) differences in the actual evoked responses or (ii) a cross-talk between the impulse-evoked response and ongoing brain activities that preserve the pre-stimulus phase and become superimposed with the evoked response.'

This is an interesting point that with the current data might be difficult to address. The impulse stimulus initiates a phase-locked evoked response, which has been suggested to be the result of a phase reset (Makeig et al., 2002). Our data supports this notion showing relatively tight time bins at which there is a high VL. So, in different trials, the impulse can hit ongoing oscillations at any phase, yet you get this single time bin of increased VL. This is in line with a phase-reset, paired with a magnification of relative differences in amplitude among channels, leading to enhanced decoding a little while after impulse presentation. The phase reset suggests that there might be a difference in actual evoked response patterns. Our data however do not support this (as the ERPs for the two opposite phase bins do not differ), but rather in the pattern of evoked responses representing the WM content (see Supplementary Figure 5).

It is of course possible that part of the ongoing signals and their phases could have still been preserved after the impulse stimulus that they interact with the evoked responses (as suggested in option two of the reviewer). At this moment it is difficult to exclude this possibility. Our favored speculation is that the impulse stimulus is maximally effective at revealing WM content information at phases of the oscillatory processes where the neuronal representation is strongest. The impulse stimulus then phase resets those oscillatory processes that are related to WM content, which then provides the basis for enhanced read-out. Whatever the mechanism is, the impulse stimulus reveals a multivariate pattern in ongoing ERPs within specific time windows of the ERPs, in a manner that depends on the overall excitability fluctuations indexed by alpha and theta. The impulse stimulus thus reveals that phase modulates information content. We have added the two possibilities related to the evoked response dynamics in the discussion (line 259-262) and Supplementary Discussion.

3. It would be important to address the anatomical localization of these effects, preferably with source localization but at least with scalp topographies. The authors acquired 64-channel EEG but used only 17 parieto-occipital channels for decoding - why is this? What was the rationale (a priori vs. trying many combinations) for the channel selection?

We presented a visual impulse stimulus, so occipital/parietal channels were most likely to show the increase evoked responses needed for decoding. This 17 channel selection was identical to Wolff et al., (2015). To address the localization, we have 1) repeated the analysis for both frontal as well as all channels (see answer to 3c) and 2) shown scalp topographies of the ERP (Supplementary Figure 5) and the weights of the decoding (Supplementary Figure 2). We found no phase dependent decoding for either the frontal or all channels. For the ERPs we found a clear visual evoked response in which we first found positive occipital focus which switch to negative later in the trial and also spread to more frontal sites. The topographies of the weights show a clear focus on the central occipito-parietal channels.

3b. In relation to Suppl. Fig. 4, the authors claim "Instead, the distributed response profile evoked by the impulse more closely matches the response profile of the memorized item when the impulse is presented within a specific restricted phase range.": this is something that could be explicitly tested by assessing whether the similarity of the sample- and impulse-responses is phase dependent.

This question might have arisen from a misunderstanding. Originally, we investigated whether the multivariate pattern of the impulse response can be decoded from a classifier trained on the sample response. Thus this is in that sense testing "*whether the similarity of the sample- and impulse-responses is phase dependent*". We merely used the ERP analysis to confirm that we were not looking at univariate amplitude differences. We ensured to make this distinction clearer stating that "*Instead, the decoding finding can only be explained if the distributed response profile evoked by the impulse more closely matches the response profile of the memorized item when the impulse is presented within a specific restricted phase range*" (page 8, line 174-176).

3c. The conclusion "Our findings are the first to show in human parietal-occipital cortex a phase-dependent modulation of neural representation strength of memory content in the range of alpha and theta." is not only questionable with respect to frequency localization (see 1. and 2.) but also unwarranted in terms of the anatomical localization: there is no evidence that the signals yielding the decoded information were produced in the parieto-occipital cortex. These signals could be produced almost anywhere in the brain and be picked up in the authors channel-selection via volume conduction. Addressing the decodability across the entire 64-channel data and preferably with source reconstruction methods is fundamental for drawing any anatomical inferences.

We regret that we made such strong conclusion about the decodability from the occipital/parietal regions. We repeated the analysis for both the frontal channels and for the full 64-channels and did not find any significant decodability following the impulse stimulus (see Supplementary Figure 4 E+F). While this is an interesting finding, this does not preclude that the signal we pick up from the parieto-occipital channels could originate from different areas. In the paper, we ensured to not make any interpretation regarding the anatomical origin of measured effects.

4. Overall the data are rather scarcely illustrated. Instead of bar plots, there are more modern ways to illustrate the distributedness of the findings. For the behavioral results, it would be more tangible to express them in terms of hit rates and the phase dependence of the hit rates per se. The circular phase plots (Fig. 2C, Suppl. Fig. 7B) would in my opinion be better shown on a linear phase axis and by illustrating the reliability by confidence limits (surrogate data and/or bootstraps).

We hope the updated figures give a better image of the distribution of the data. Specifically, as much as possible we displayed all data points of all individual subjects giving an indication of the shape of the distribution using a spread plot (For example Figure 1B, 2A-C). Figure 2B provides the phase dependence of the (weighted) hit rates showing the hit rates for the two different phase bins (as well

as for the noImpulse trials). We replaced the circular phase plots by a linear phase axis with confidence intervals. The comparison with the bootstrap (in which we tested the absolute phase distance from 0) is displayed in Figure 2D.

Reviewer #3 (Remarks to the Author):

The present manuscript, titled “Phase-dependent enhancement of working memory content and performance,” follows on previous work showing (1) that the content of working memory can be decoded from EEG data based on the response to high-intensity impulse stimulus presented during the delay between the sample item and the target item (i.e., while the sample is being held in working memory) and (2) that working memory is linked to low-frequency oscillations. Their results suggest that the distributed neural activity associated with memorized information waxes and wanes with the phase of these low-frequency oscillations. That is, classification of the sample item when an impulse stimulus was presented during the memory delay (based on a linear classifier) was more accurate at specific phases of neural activity in the theta/alpha range. Their results also suggest that behavioral performance (i.e., behavioral accuracy) was better on trials when the impulse was presented during phases

associated with better decoder accuracy. Their results thus indicate that impulse stimulation has the potential to improve working memory, if the timing of the presentation is properly aligned with underlying oscillatory activity. These are certainly exciting conclusions, and the authors took the commendable step of replicating their results in a second dataset. However, there are several issues/concerns that I would like the authors to comment/address before I can be convinced of their results.

1. In the introduction (page 3): “...the memorized information can be decoded from EEG electrodes over occipital and parietal lobes.” It would be helpful to add a sentence or two here about why these might have more information than frontal electrodes. Also, the authors might consider removing the location of the electrodes from the abstract, as there’s no explanation there for why they restricted recordings to occipital and parietal regions.

We have added the information why we choose the specific electrodes. We presented a visual impulse stimulus, so occipital/parietal channels were most likely to show the increase evoked responses needed for decoding. This channel selection was identical to Wolff et al., (2015). To address the issue of localization further, we 1) repeated the analysis for both frontal as well as all channels and 2) now show scalp topographies of the ERP (Supplementary Figure 5) and the weights of the decoding (Supplementary Figure 2). We did not find any significant effect using either the frontal or all channels. Still it is impossible to claim the effect to originate from the occipital cortex using channel EEG, so we ensured to not make any conclusions regarding this topic.

2. In the introduction (page 3): “In addition, given the role of low frequency oscillations in working memory and existing theoretical proposals...” This is vague. Can you be more specific regarding these theoretical proposals?

We regret this was vague. We aimed to refer back to the second paragraph in which we explain proposals suggesting that the phase of low frequency oscillations can represent working memory content (*“it has been suggested that information is most strongly represented at restricted phases for which excitability levels are high”*). To make this more clear we now summarized stating that *“Given the proposed theoretical framework suggesting that the phase of low frequency oscillations represents working memory content^{9,25}, we additionally expected...”* (page 3, lines 56-58).

3. In the results (page 5): “...we first estimated phase at impulse onset... using an FFT approach including three cycles of data prior to impulse onset.” Please clarify. For each frequency, you used data equivalent to three cycles, meaning the amount of data for each frequency was different? Also, an FFT provides the frequency content of a signal across multiple frequencies, unlike, e.g., a wavelet, which is centered on a specific frequency. Why were multiple FFTs run here?

For a good phase estimation, you have to find a balance between providing a wide enough window to reliably extract oscillatory features as well as providing a small enough window so that it is likely that within the extracted window the phase does not shift (i.e. the oscillation should be stationary with the analysis window). For higher frequencies this stationary is unlikely to hold over many hundreds of milliseconds. Therefore, we wanted to extract frequency content using the same amount of cycles per frequency (in our case 3 cycles). To ensure this we needed to run multiple FFTs for each frequency point separately (including 3 cycles of data for the specific frequency point). Indeed, an alternative would have been to do multiple wavelet analyses in which implicitly less data is used for higher frequencies. We have added a shortened version of this explanation in the method section (page 15, line 387-388).

4. Figure 1A. Here, the authors present the phase-specific results (i.e., whether there was phase consistency prior to the impulse stimulus on correctly classified trials). It was previously stated that the classifier could identify the orientation of the working memory item in 12 ms data bins collected 0-150 ms following the impulse stimulus. Here, the figure shows 0-250 ms, with the significant VL for time points near or after the offset of the impulse stimulus. What was the corresponding, non-phase dependent classifier accuracy at each time point? Second, is it possible to get corresponding VLs for the incorrectly classified trials, and what is the difference in VA between correctly and incorrectly classified trials? Finally can the authors speculate at all regarding the timing of these effects relative to phase measurements prior to impulse stimulation (i.e., 150 ms earlier)? The results presented in Supplemental Figures 6 and 7, show significant results in the 150ms period corresponding with impulse stimulation.

We regret the typo, we classified from 0-250 ms. Please find the phase + non-phase dependent temporal evolution of the classifier in Supplementary Figure 2B and below. As you can see is that for specific time windows the VA shows a significantly higher decoding compared to including all data.

Additionally, we ran the VL analysis for incorrectly classified trials. No clear phase consistency was found for these trials (lowest p-value of cluster was 0.575; see vector length difference below).

To clarify the 0-250ms time interval chosen for analysis: A priori it was expected that the impulse stimulus would phase-dependently activate the sensory representation of the maintained item within a time window following the impulse stimulus within which sensory modulation could be expected. Therefore, latencies within the window of sensory evoked responses (50-250 ms) were expected to be modulated by the phase of the stimulus. Indeed, in both datasets we found effect within these time windows. But in the dataset of Wolff et al this effect was earlier. This difference could be due to many factors. In our data the onset of the impulse stimulus was more jittered, which might influence the exact timing of the sensory response (added to the supplementary discussion and also discussed related to the comment of reviewer 2, point X). Secondly, our task was more difficult as we also included trials with lower angle differences. Finally, we tried to fully replicate the study based on all the information in the paper, but we can never ensure that all details were identical (lighting conditions etc). Combined, this could have led to a different latency between the two studies. However, both latencies are within the expected sensory range (see also our response to point 1b of reviewer 2).

5. Figure 1B. What frequency and time point are being represented here?

It represents per participant the frequency and time point at the maximum VL-value of the significant clusters. We regret it was not clear and added it into the figure legend.

6. Are there differences in oscillatory power just prior to impulse stimulation that might also influence classifier accuracy?

We correlated the log power with the classifier accuracies. For this analysis no significant effects were found (lowest p-value was 0.846).

7. Perhaps the biggest weakness reported in the results is that the phase of oscillatory activity associated with better classifier and behavioral accuracy differs across participants. The authors speculate that this could be related to methodological disadvantages of EEG, such as volume conduction and the superposition of electric potentials, but I don't find this explanation satisfying. Are the authors suggesting that they are detecting different memory-related signals across participants that are all predictive of classifier and behavioral accuracy? If these effects trace back to a single neural basis, then how would volume conduction or the superposition of electrode potentials lead to subject-specific phases? Although I'm not satisfied with the explanation, my concerns are somewhat mitigated by the subject level convergence for the phases related to better classifier and behavioral accuracy.

We thank the reviewer for his/her remark. Indeed, the phases were not consistent over subjects. There are two possible explanations:

- 1) As we have already suggested in the paper it could be that due to the superposition of the electrical potentials the phases differ over participants. This could be a consequence of different anatomy of the participants leading to different folding patterns, in turn leading to differences in polarity of the electric field. Moreover, the exact anatomical location where the representation is held could differ over participants due both to neurophysiological reasons (different participants may represent the stimulus in a stronger or weaker fashion in different parts of the occipital cortex), and due to anatomical reasons (differences in sulcal patterns as well as differences in cortical magnification and retinotopy in the different areas among participants), leading to a different phase as measured on the scalp. There have been previous reports in which the phase of the EEG was not consistent over participants (see e.g. Henry & Obleser, 2012).
- 2) The actual phase which represents the information might also differ over participants. While it is clear that the phase at which neurons have the highest firing rates and gamma power at rest is consistent, it is not evident why these should be also the phases that contain the most information. This has previously not been tested and we are among the first to extract the information instead of the activity patterns themselves.

Finally, as the reviewer also mentions, we are confident that the phase we are extracting (at the moment of impulse stimulus presentation) is behaviorally relevant considering that the phase of the optimal classification and behavior is similar. The reasons why the phases are not consistent over participants was added to the manuscript: "*This may be due to the fact that in EEG signal, the phase of oscillatory signals measured at the scalp is determined by many factors, for example volume conduction*

and the superposition of electric potentials, that could reduce this phase consistency over participants. Alternatively, the exact phase at which the most working memory information (in contrast to the most activation) is present is not identical over participants” (page 6-7, line 135-140).

8. In the results (page 7): “We tested the observed classifier accuracy to the median of the chance distributions of accuracies obtained after 1000 permutations of the correct/incorrect classification labels at the stage of calculating the VL.” To test whether the observed VL is higher than that expected by chance, shouldn’t you see if it exceeds 950 (for $p < 0.05$) of the values in the test distribution (i.e., the randomly permuted distribution)? Instead the authors use the median from this permutation approach as a benchmark for further statistical testing. Please explain. These same questions/concerns apply to the analysis that examines pre-classifier phase and behavioral accuracy.

The analysis follows commonly used methods for random effect analysis using multivariate pattern analysis. It is common to extract the classification which is expected based on chance. This ‘empirical’ chance level is created by calculating a null distribution for every participants and comparing the observed accuracies with this empirical level (see e.g. Formisano et al., 2008). The median reflects the 50th percentile for this individual distribution, capturing the central tendency of this distribution, thereby reliably reflecting empirical chance. In this way, random effect conclusion can be drawn which are better generalizable to the population than creating a single null distribution over participants. We repeated the same approach for both the VL and the behavioral analysis in which for each participant and data point we extracted the value that would be expected on chance.

9. Figure 2. The figure doesn’t match the caption. The caption refers to (D), which seems to be (B), and there is no (D).

We apologize for this mistake and corrected the caption references in figure and figure legend.

10. In the methods (page 13), there’s a typo: “long impulse trials, short no-impulse trials, long impulse trials.” Should be: long impulse trials, short no-impulse trials, long no-impulse trials.

Thanks for noticing this, we have adjusted this typo.

11. In discussion (page 11): “Our results are also of direct relevance for the discussion where in the brain working memory content is maintained...” I don’t think this (and the paragraph that follows) is a fair statement, since it seems the authors only examined electrodes over occipital and parietal regions.

We agree that we cannot make directly conclusion on localizations and removed any reference to this.

Reviewers' Comments:

Reviewer #1:

Remarks to the Author:

The authors have satisfactorily addressed the points that I raised in my initial review.

Reviewer #2:

Remarks to the Author:

The authors have improved the manuscript in many places. The concern of the reliability of the primary observation was, however, not dealt with rigorously.

My core concern for this manuscript was the reliability of the main result, with inferential statistics illustrated in Fig. 1A. The authors argue that the data by Wolff support this result, but observing completely different latency ranges there while the evoked responses are similar does not inspire confidence. On this account, the authors argue that the difference would be explainable by earlier evoked response in Wolff. Looking at the evoked responses, however, it is clear that the author's components are on the falling and rising slopes of the negative deflection peaking at 180 ms whereas the significant clusters in the Wolff dataset are at the onset and peak of the early positive ERP component peaking at around 100 ms rather than around the subsequent, slightly earlier, negative deflection corresponding to authors' 180 ms component. At face value, the Wolff dataset seems more to challenge than support the authors' findings.

The authors also performed a split-trials reliability analysis but instead of reproducing two analyses for the main inferential statistic (Fig. 1A), and assessing their reproducibility, they confined the reliability assessment to comparing within and out-of cluster effects. As far as I see, this is somewhat akin to just a post hoc test. Moreover, the reliability value even in this approach was low. The author's analysis thus appears underpowered and bordering the significance criteria. This guess is supported by the analyses shown in Suppl Fig 4C-F, where very similar effect sizes in terms of vector length difference values and apparent cluster sizes are observed but found insignificant.

As far as I see, consolidation of the main findings with a replication cohort is the only and best way forward.

Minor

The formulation of the main result appears unsubstantiated by the data, for example Abstract line 23 "working memory information is maximized within limited phase ranges": the authors did little to clarify the physiological basis for why and how this WM content information is decodable from a narrow latency range over the impulse-stimulus-evoked response. There are no analyses explaining whether this relates to evoked activity, superposition of the evoked response and continued ongoing oscillations, or phase resetting of these oscillations, or something else. Before the sources of decodable information are elucidated, it is premature to make conclusions about "working memory information" per se and, above all, its relationship with phases of ongoing oscillations.

Prior art in decoding working memory contents could be cited more comprehensively, for example Bae, G. Y., & Luck, S. J. (2018). Dissociable Decoding of Working Memory and Spatial Attention from EEG Oscillations and Sustained Potentials. *The Journal of Neuroscience*, 38, 409-422.

Reviewer #3:

Remarks to the Author:

I am satisfied with the authors' responses. I have no further comments/criticisms.

Please find a point-by-point answer to the concerns of the reviewers. All changes in the manuscript are highlighted in yellow.

Reviewer #1 (Remarks to the Author):

The authors have satisfactorily addressed the points that I raised in my initial review.

Reviewer #2 (Remarks to the Author):

The authors have improved the manuscript in many places. The concern of the reliability of the primary observation was, however, not dealt with rigorously.

My core concern for this manuscript was the reliability of the main result, with inferential statistics illustrated in Fig. 1A. The authors argue that the data by Wolff support this result, but observing completely different latency ranges there while the evoked responses are similar does not inspire confidence. On this account, the authors argue that the difference would be explainable by earlier evoked response in Wolff. Looking at the evoked responses, however, it is clear that the author's components are on the falling and rising slopes of the negative deflection peaking at 180 ms whereas the significant clusters in the Wolff dataset are at the onset and peak of the early positive ERP component peaking at around 100 ms rather than around the subsequent, slightly earlier, negative deflection corresponding to authors' 180 ms component. At face value, the Wolff dataset seems more to challenge than support the authors' findings.

The authors also performed a split-trials reliability analysis but instead of reproducing two analyses for the main inferential statistic (Fig. 1A), and assessing their reproducibility, they confined the reliability assessment to comparing within and out-of cluster effects. As far as I see, this is somewhat akin to just a post hoc test. Moreover, the reliability value even in this approach was low. The author's analysis thus appears underpowered and bordering the significance criteria. This guess is supported by the analyses shown in Suppl Fig 4C-F, where very similar effect sizes in terms of vector length difference values and apparent cluster sizes are observed but found insignificant.

As far as I see, consolidation of the main findings with a replication cohort is the only and best way forward.

We thank the reviewer for his/her comments. We acknowledge that the reviewer still perceives our findings as inconsistent with the sample of Wolff et al., in particular with respect to the differences in the timing of the significant effects. We want to stress once more that our hypotheses and conclusions are not concerned with the exact timing of our significant clusters. We neither had a priori expectations related to the timing of significant clusters, nor do we have any conclusions that depend on the exact timing of the cluster effects. Instead, our focus was on the phase-dependency of working memory representations.

We suggest that with respect to the for us irrelevant timing of significant clusters, one needs to be prudent in directly comparing the datasets. Besides the previously mentioned differences in ERPs that could explain latency differences, there are also statistical considerations that make a simple comparison of apparent differences between the timing of significant effects in the two cohorts difficult. Data gathered from two independent cohorts will never be identical and it can be deceiving to directly compare p-values from the two sets. Indeed, non-significant effects in a replication do not directly discredit the effect in another dataset (Simonsohn, 2015; Gelman & Stern, 2006; Nieuwenhuis, Forstmann, & Wagenmakers, 2011). Thus, the claim that the Wolff dataset is challenging our findings (or that our data set would challenge that of Wolff) can in fact not be made without performing a proper statistical test.

To address this question, we asked the editor for advice, as a third replication is unlikely to resolve this issue. Following the editor's suggestion, and after consulting further with statistical experts in our faculty (Dr. Giancarlo Valente and Dr. Alberto Cassese), we now considered that comparing only the significant results from the two datasets is not necessarily the best way to assess replication success when statistical power is limited. We therefore conducted additional analysis to directly address this issue and to determine whether there is evidence that the two results are statistically inconsistent, following the approach described by Simonsohn (2015).

In this approach, one investigates whether the effect size in the replication cohort is statistically different from an effect size at which an effect was *detectable* in the original dataset. This constitutes a test of replicability (Figure 1). We followed Simonsohn's (2015) proposal to investigate effect sizes at a power of 33% (i.e. a small effect) in the original dataset (cohort 1, our dataset). Effect sizes were estimated using Cohen's D (the standardized difference between means). Only sporadic datapoints at the edge of the cluster, indicated in yellow in the figure below (added as Supplementary Figure 9), showed Cohen's D in cohort 2 that were significantly lower than detectable at an effect size of 33% in cohort 1. Reversing the order of the test (comparing Cohen's D of cohort 1 at an effect size at 33% power detectable in cohort 2) did not yield any point at which Cohen's D was lower within the clusters of cohort 2. Therefore, on a datapoint-by-datapoint basis there was no evidence that the data in the two datasets are *inconsistent* with each other.

Figure 1. Evaluation of replication results. A) Cohen's D for cohort 1 (our data). B) Cohen's D for cohort 2 (Wolff's data). C) Statistical power comparisons. Different colors indicate the following: *Dark blue*: Cohen's D of cohort 2 was not significantly lower than what would be detected at a power of 33% in cohort 1 outside of the clusters of cohort 1 ($D2 > 33\% \text{pow}$). *Light blue*: Cohen's D of cohort 2 is significantly lower than what would be detected at a power of 33% in cohort 1 outside of the cluster of cohort 1 ($D2 < 33\% \text{pow}$). *Green*: Cohen's D of cohort 2 was not significantly lower than what would be detected at a power of 33% in cohort 1 and is within the cluster of cohort 1 ($D2 > 33\% \text{pow} \ \& \ \text{clust1}$). *Yellow*: Cohen's D of cohort 1 was significantly lower than what would be detected at a power of 33% in cohort 1 and is within the cluster of cohort 1 ($D2 < 33\% \text{pow} \ \& \ \text{clust1}$). Note that light blue points were never significant in the original dataset, and therefore were irrelevant for judging the replication success for a significant effect. D) Same as C but investigating whether there was evidence that Cohen's D of cohort 1 was lower than what would be detected at a power of 33% at cohort 2.

As we did not find any evidence on a datapoint-by-datapoint basis that the two cohorts are inconsistent, we went further to now validate and compare our *a-priori* defined statistical test for the two cohorts to further support the consistency of the two datasets. This statistical test reflects the cluster statistics in which we asked the question whether there is any evidence that theta/alpha phase modulates decoding performance within the sensory processing temporal window (see e.g. line 67-70 of main paper). Note that from the test-statistics we used to address this question, *no inferences* can be made about the *exact* timing or frequency (Maris et al., 2000; Sassenhagen & Draschkow, 2019). However, the test statistics used provides a powerful approach for finding a significant effect in a specified time/frequency *window* of interest. In both cohorts we consistently find a significant effect of phase modulation in the time window of interest ($p = 0.016$ and $p = 0.035$ for the strongest cluster in cohort 1 and cohort 2 respectively). The significant findings in both cohorts already indicates the consistency in the datasets.

Another approach to assess reproducibility of our main inferential statistic is to extract the probability under the null hypothesis of no cluster in the data given the information of both datasets. As such, we can create a new test statistic: the average of the cluster statistics from both cohorts. The null distribution under this null hypothesis is constructed by creating a null distribution of the average of the random permutations. The p-value can then be extracted by the number of permutations that have a cluster statistic higher than this observed value (expressed as a proportion of the total number of permutations). The p-value in this approach is 0.007, thus rejecting the null-hypothesis that for the combined data there would be no significant cluster.

Figure 2. Histogram of the null distribution for the average over two cohorts. Dotted and full red line indicate the 95th percentile and observed average cluster statistic.

We believe that these new and additional analyses directly address the concern of the reviewer that we failed to compare the two datasets directly with each other. The results of the presented statistical analyses show no evidence of inconsistency between the two datasets in terms of the timing of clusters, and they confirm the consistency of the effects of interest in our study (the phase-dependent decoding of WM content). In the light of the results of our additional analyses, we suggest therefore there is no reason to provide a third dataset. We nevertheless appreciated this discussion as it allowed us to provide further evidence for the consistency of our own dataset and that of Wolff et al. Besides the previously reported split-half approach, we now included the two additional approaches in our supplementary methods.

Minor

The formulation of the main result appears unsubstantiated by the data, for example Abstract line 23 “working memory information is maximized within limited phase ranges”: the authors did little to clarify the physiological basis for why and how this WM content information is decodable from a narrow latency range over the impulse-stimulus-evoked response. There are

no analyses explaining whether this relates to evoked activity, superposition of the evoked response and continued ongoing oscillations, or phase resetting of these oscillations, or something else. Before the sources of decodable information are elucidated, it is premature to make conclusions about “working memory information” per se and, above all, its relationship with phases of ongoing oscillations.

We regret the unclear formulations that the reviewer mentions and like to clarify our interpretation. We interpret any phase effect as *relative to the phase of ongoing oscillations* as we extract the phase from a time period prior to any evoked response, i.e. prior to the impulse stimulus (see line numbers 95-100 and line numbers 386-387). Sorting the decoding results of the WM content based on these phases shows increased decoding for specific phase ranges. We can exclude the possibility that this effect is due to changes in signal-to-noise ratio as the ERPs between the best and worse decoding phase don't differ (see line numbers 168-176 and Supplementary Figure 5). Therefore, the nature of this improved decoding is physiological.

We believe what we state above is in full agreement with the reviewers' comment. However, since we decode information related to WM content (as the labels of the classifier reflect WM orientation and the behavioral findings confirm that we extract WM behaviorally relevant information), any conclusions that will show a difference in decoding will automatically relate to WM information (i.e., we model WM information, hence the conclusions will be about WM information). As we can show that the improved decoding is at least physiological (and not artifactual), we respectfully disagree with the notion of the reviewer that '*the sources of decodable information*' need to be known before one can conclude anything about WM information. We can conclude that there is a physiological increase in WM information at specific phase ranges, although we grant it is not known which physiological mechanism enables this.

At this point it is difficult to extract the exact physiological basis of increased decoding at specific phase ranges (evoked activity, phase reset, superposition of evoked response + ongoing oscillations). We mention this in the Supplementary Discussion. We changed this text as we realized it might still contain unclarities. It now reads: *The exact origin of the phase-dependent distributed response to the impulse stimulus is at present unclear, but we can conceive of two broad categories of mechanisms. On the one hand, the phase-dependent distributed response could result from differences in the evoked response among electrodes, or, alternatively, it could result from a cross-talk between the impulse-evoked responses at the different electrodes and the ongoing oscillatory brain activity. According to the former idea, from trial to trial, the impulse will hit ongoing oscillations and induce a phase-reset, aligning all phases to one phase (i.e. the post-impulse decoding is independent from ongoing oscillatory activity). For impulse presentations at specific phase ranges, the evoked responses would more clearly resemble the evoked response elicited by the original sample stimulus, resulting in an increased VL. One mechanism by which this might occur is that, too the extent that WM*

content is encoded in small differences in activity of neuronal population at specific phase ranges, the strong phase-reset would momentarily reveal these amplitude differences among neuronal populations more clearly, and make it possible to read them out from differences in EEG scalp topographies. Thus, the phase-reset is in this case paired with a magnification of relative differences in amplitude among channels, leading to enhanced decoding of WM content a little while after impulse presentation. Alternatively, according to the latter idea, it is possible that part of the ongoing oscillatory signals and their phases are preserved after the impulse stimulus so that they interact with the evoked responses, in a manner that manifests itself in phase-dependent decoding of WM content following the impulse stimulus. While we did not find any evidence for increased decoding at specific phase ranges in the retention period prior to the impulse, it is still possible that the preservation of ongoing oscillations after the impulse stimulus adds to the decoding performance. At this moment it is difficult to decide among these and likely also other possible mechanisms.

Prior art in decoding working memory contents could be cited more comprehensively, for example Bae, G. Y., & Luck, S. J. (2018). Dissociable Decoding of Working Memory and Spatial Attention from EEG Oscillations and Sustained Potentials. *The Journal of Neuroscience*, 38, 409-422.

We regret not citing relevant decoding WM literature and added this throughout the manuscript (see e.g. line numbers 38-40 and line numbers 247-248).

Reviewer #3 (Remarks to the Author):

I am satisfied with the authors' responses. I have no further comments/criticisms.

Reviewers' Comments:

Reviewer #1:

None

Reviewer #2:

Remarks to the Author:

The authors have adequately considered the comments so far. In my opinion, a replication cohort would have consolidated the paper much better than yet additional statistics on the existing data. As the authors point out, this also is a non-trivial issue and a replication cohort might not yield the desired result either because of limited power. Nevertheless, in a pooled analysis, additional subjects would still yield increased power.

What I see problematic here is that the authors acknowledge that they have a low-powered study and yet, even for a high-impact publication, choose to not double the N in order to increase the power and ascertain that the result holds. EEG recordings and scalp-electrode level analyses are not hugely time consuming and healthy controls are easy to recruit. (There can be more unique datasets, say involving patients, primates, invasive recordings, etc., where low N is more justifiable.) Hence accepting or rejecting this approach is a matter of editorial policy and, in my opinion, a question of risk management - the result is very interesting if true but of little value if it is later found to be not replicable.

I have no further comments on this and see this as a matter for the Editor to evaluate.

Reviewer #3:

Remarks to the Author:

With regard to the reproducibility of their results across the two datasets, I am satisfied with the authors' response, including additional analyses.